# Diversity of Ladybird Beetles (Coleoptera: Coccinellidae) in Tenerife and La Gomera (Canary Islands): The Role of Size and Other Island Characteristics

**DOI:** 10.3390/insects15080596

**Published:** 2024-08-06

**Authors:** Jerzy Romanowski, Piotr Ceryngier, Jaroslav Vĕtrovec, Christian Zmuda, Karol Szawaryn

**Affiliations:** 1Institute of Biological Sciences, Cardinal Stefan Wyszyński University, Wóycickiego 1/3, 01-938 Warsaw, Poland; p.ceryngier@uksw.edu.pl (P.C.);; 2Buzulucká 1105, 50-003 Hradec Králové, Czech Republic; jerryvetrak@seznam.cz; 3Museum and Institute of Zoology, Polish Academy of Sciences, Twarda 51/55, 00-818 Warsaw, Poland; k.szawaryn@gmail.com

**Keywords:** biodiversity, Canary Islands, alien species, island biogeography

## Abstract

**Simple Summary:**

The main assumption of the so-called theory of island biogeography is that larger islands are home to more species than smaller ones. However, species richness on an island is affected not only by its size but also by other features, such as the island’s distance from the mainland or its geological age. The two Canary Islands Tenerife and La Gomera are a good model for testing the relationship between species richness and island area, as they differ considerably in size (the former has an area 5.5 times that of the latter) but are similar in location and age. They lie close to each other (less than 30 km apart) in the central-western part of the archipelago, about 300 km from the African coast. Both were formed as a result of volcanic activity some 11.5–12 million years ago. We compared the species composition and species richness of ladybird beetles on these two islands based on our field surveys and the literature data. As expected, clearly more ladybird species have been recorded on Tenerife (47 species) than on La Gomera (26 species). Being the largest of the Canary Islands, Tenerife has the richest ladybird fauna of all the islands in the archipelago, but it is also more susceptible to colonization by non-native ladybirds than the other islands: to date, ten species of non-native ladybirds have been recorded on Tenerife, compared to between three and seven on the other islands (five on La Gomera). Among the non-native ladybirds established on Tenerife (but not on any other island in the archipelago) is the harlequin ladybird (*Harmonia axyridis*), a highly invasive species of Asiatic origin that has spread nearly worldwide.

**Abstract:**

This paper provides new data on the ladybird beetles (Coccinellidae) from two islands in the Canary archipelago: Tenerife, the largest island, and La Gomera, the second smallest. As they clearly differ in size but are similar in location and geological age, they are a suitable model for testing the species–area relationship. Our study shows that, in line with this main assumption of the theory of island biogeography, clearly more species occur on a large island (Tenerife) than on a small one (La Gomera). The field surveys documented the occurrence of 35 ladybird species on Tenerife (including 5 not previously reported from this island) and of 20 species on La Gomera (2 species new to the island). *Coelopterus* sp. collected on Tenerife (a single female that could not be identified to species) is the first record of this genus for the whole Canary Islands. Taking our data and previously published records into account, 47 species of Coccinellidae are known to occur on Tenerife and 26 species on La Gomera. Tenerife has by far the richest ladybird fauna of all the Canary Islands (the next in line, Gran Canaria, has 41 recorded species), but it also has the highest number of non-native ladybird species. All of the ten non-native species recorded in the Canary Islands are found on Tenerife, and for most of them, Tenerife was the island of their first appearance in the archipelago. This island, much more distant from the mainland than the other relatively large islands (Fuerteventura, Lanzarote), appears to be the main recipient of ladybirds immigrating to the Canary Islands. Tenerife can play this role probably because of its great habitat diversity and altitude variation, as well as intensive tourism and trade-related transport.

## 1. Introduction

The Canary Islands, along with four other Atlantic archipelagos of volcanic origin, the Azores, Madeira, the Salvages and Cape Verde, belong to the biogeographical region known as Macaronesia. The region as a whole, and the Canary Islands in particular, are characterized by high rates of endemism in many taxonomic groups, especially vascular plants, land snails and arthropods [1]. Of about 7300 native species of terrestrial arthropods recorded in the Canary Islands, 44% are endemics (see Supplementary Material in [1]. A substantial part of them belong to the insect order Coleoptera (beetles). Oromí et al. [2] list about 2150 species and subspecies of beetles known to occur in the Canary Islands, of which 1376 (64%) are marked as endemics.

The systematic study of Coleoptera of the Canary Islands was initiated in the 19th century by Wollaston [3,4]. In addition to many other beetles, he identified 15 species of ladybirds (Coccinellidae) inhabiting the archipelago, most of which turned out to be Canarian or Macaronesian endemics. Two of the reported species (*Scymnus oblongior* and *S. maculosus*) turned out to be synonyms of another Wollaston’s species, *Nephus flavopictus*, previously described from Madeira, and one variety (*S. canariensis* var. *rufinennis*) is now considered a distinct species. The number of Coccinellidae species reported by Wollaston and the number of species currently distinguished in his material is the same (15), despite some differences between the two sets. Later researchers [5,6,7,8,9,10] recognized many other ladybird species there, including several endemics.

In recent years, the highly unique fauna of Coccinellidae of the Canary Islands has been subjected to updating surveys carried out by our team [11,12,13,14,15]. The most important outcomes of these surveys can be summarized as follows: (1) many species new to each of the islands studied were recorded, indicating that the ladybird fauna of individual islands has not been thoroughly investigated; (2) various geographical and historical elements, such as the Canarian and Macaronesian endemics, species common to the Canary Islands and neighbouring areas of NW Africa and/or SW Europe or alien species from remote regions, could be distinguished in the Canarian ladybird communities; and (3) the Canary Islands have been subject to frequent colonization by ladybirds of various origins.

So far, we published data on Coccinellidae of five of the seven main islands of the Canarian archipelago: Fuerteventura [11], Lanzarote [12], Gran Canaria [13], El Hierro [14] and La Palma [15]. In this paper, we provide information on ladybirds of the remaining two islands, Tenerife and La Gomera. These islands of similar geological age (11.5–12 Myr) [16], located in the central-western part of the archipelago, are less than 30 km from each other and about 300 km from the African mainland. On the other hand, they clearly differ in size: Tenerife, measuring 2034 km^2^, is the largest of the Canary Islands, while La Gomera (370 km^2^) is the second smallest island [16]. These similarities and differences make Tenerife and La Gomera a suitable model to test one of the central assumptions of the theory of island biogeography, the species–area relationship, according to which the species richness in an island is positively related to the island’s size [17].

## 2. Study Area, Materials and Methods

Due to their volcanic origin, both Tenerife and La Gomera are eminently mountainous. Tenerife, not only the largest but also by far the highest of the Canary Islands, reaches its culmination (the peak of Teide) at 3718 m a.s.l., while the highest point of La Gomera, Alto de Garajonay, rises to 1487 m a.s.l. The mild subtropical Atlantic climate of the islands is strongly influenced by the humid trade winds that transport moisture from the northeast. As a result, the northern and eastern areas experience relatively humid conditions, while those in the south and west receive low and infrequent precipitation. The mountainous nature of the islands, combined with the climate-shaping trade winds, results in a specific zonation of forest and non-forest vegetation. One of the most unusual Canarian plant communities, the laurel forest (laurisilva), occupies medium altitudes in the north and east of the islands. This relict subtropical rainforest, characterized by lush evergreen vegetation, is closely dependent on the moisture brought by the trade winds. Other important habitats in Tenerife and La Gomera include lowland semi-desert scrub vegetation with *Euphorbia* spp., higher altitude pine forests dominated by the Canary Island pine (*Pinus canariensis* C. Smith) and montane shrublands above pine forests [16,18]. Important for our ladybird investigations were also ornamental plants grown in towns and villages, along roadsides, in parks, etc. These are often infested with coccids, aphids, aleyrodids and other soft-bodied arthropods, which are prey of many species of Coccinellidae, especially of non-native origin.

Ladybirds were collected in all the habitats described above, and from various plants at 57 sites on Tenerife and 30 sites on La Gomera (Figure 1, Appendix A) using standard methods, such as shaking insects from trees and shrubs on a 1 m × 1 m beating tray, sweeping low vegetation with a net or picking directly observed specimens. Each individual was noted, even if some were released after in situ identification. The species were identified based on the morphological and anatomical details documented in our earlier papers [11,13,14,15]. Below, in the Results section, we provide the details of the records (the island’s name, locality, date, number of adults, number of larvae and pupa (when recorded), habitat and/or host plant) for each of the recorded species, which are systematically ordered according to the phylogenetic analysis made by Che et al. [19]. Most of the material was collected by J.V. (10.XII–17.XII.2016, 25.XII.2017, 10.I–19.I.2020, 30.I–7.II.2020, 20.XI–30.XI.2021, 30.I–6.II.2023) and J.R and P.C. (8.IX–11.IX.2019, 15.VI.2021, 22.VI.2021, 23.II–1.III.2022). If the specimens were collected by other people, their names are indicated after the record details. The voucher specimens are stored in the insect collection in the Institute of Biological Sciences, Cardinal Stefan Wyszyński, University in Warsaw, and in the private collection of Jaroslav Větrovec. The morphological and anatomical details of several species were photographed using an Olympus DP23 digital camera attached to an Olympus BX43F compound microscope. Simple linear regression was used to determine the relationship between the number of ladybird species and the area of the Canary Islands. The data were analysed for normality using the Shapiro–Wilk test. All statistical analyses were performed using Statistica 12 (StatSoft, Inc., Tulsa, OK, USA).

## 3. Results

Altogether, 2368 ladybird individuals were recorded in this study, of which 1560 individuals (1531 adults and 29 larvae) belonging to 35 species (including 1 identified to the genus level) were found on Tenerife and 808 individuals (674 adults, 92 larvae and 42 pupae) belonging to 20 species on La Gomera. Detailed data on all the recorded species are provided below.

Microweiseinae Leng, 1920Serangiini Pope, 1962


***Delphastus catalinae* (Horn, 1895)**


**Tenerife.** Afur env.: 4.II.2017, 1 ex.; Barranco de Acetenjo: 25.XII. 2017, 1 ex.; Arico, El Sauzal, Los Pinos, Puerto de la Cruz, Santiago del Teide, Tamaimo, Tegueste: 08.IX–11.IX.2019, total of 31 exx. collected mostly from *N. oleander*, *Hibiscus* sp., *Ficus* sp. and *Prunus dulcis* (Mill.) D.A.Webb; Puerto de la Cruz: 22.XI.2021, 3 exx.; Santa Cruz De Tenerife: 23.II.2022, a total of 38 exx. collected mostly from *Bambusa* sp., palm trees and *Ficus* sp.; Aeropuerto de Tenerife Sur: 01.III.2022, 8 exx. from *Ficus* sp.; Las Maretas: 6.II.2023, 80 exx.

**La Gomera.** Playa Santiago, San Sebastián de La Gomera, Santa Ana, Valle Gran Rey, Vallehermoso: 24.II–27.II.2022, a total of 119 specimens collected mostly from *Ficus* sp. and *Nerium oleander* L.

**Remark.** This species is alien to the Canary Islands. Its native range includes Colombia, Mexico, southern California (USA) and the island of Trinidad [20].

Coccinellinae Latreille, 1807Stethorini Dobzhansky, 1924


***Stethorus tenerifensis Fürsch*, 1987**


**Tenerife.** Adeje: 1.V.1994, 3 exx. (leg. A. Machado); Area Recreativa Las Lajas, Chimiche, El Sauzal, La Esperanza, Las Galletas, Puerto de la Cruz, Santiago del Teide, Tamaimo, Tegueste, Vilaflor: 08.IX–11.IX.2019, total of 49 exx. collected mostly from *Pinus canariensis* C. Smith, *N. oleander*, *Hibiscus* sp., *Ficus* sp., palm trees and *Prunus dulcis*; Aeropuerto de Tenerife Sur: 22.VI.2021, 24 exx. collected mostly from *N. oleander*, and *Hibiscus* sp.; Montes los Frailes: 27.XI.2021, 1 ex. from *Pinus canariensis*; Santa Cruz De Tenerife: 23.II.2022, 16 exx.

**La Gomera.** Agulo, Alajeró, San Sebastián de La Gomera, Valle Gran Rey, Vallehermoso: 24.II–27.II.2022, a total of 42 exx. collected from various plants including *N. oleander*, *Ficus* sp., *Casuarina equisetifolia* L. and *Ricinus communis* L.

***Stethorus wollastoni*** **Kapur, 1948**

**Tenerife.** Barranco de Acetenjo: 23.XII. 2017, 5 exx. (det. I. Kovář), 25.XII.2017, 1 ex. (det. I. Kovář).

**La Gomera.** El Cedro: 5.X.2008, 2 exx. (leg. P. Stüben); Cerco de Armas: 11.I.2020, 9 exx.; Laguna Grande: 28.II.2022, 6 exx. collected from Lauraceae, and 1 ex. from *Argyranthemum broussonetii* (Pers.) Humphries.

Coccinellini Latreille, 1807


***Adalia decempunctata* (Linnaeus, 1758)**


**Tenerife.** Arona: 10.IX.2019, 1 ex. from *N. oleander.*


***Cheilomenes propinqua* (Mulsant, 1850)**


**Tenerife.** Los Christianos: 06.VI.2022, 3 exx. from *Coccoloba uvifera* L., (leg. A. Krzysztofiak).


***Coccinella miranda* Wollaston, 1864**


**Tenerife.** El Rosario: 20.I.2013, 5 exx.; Aguamansa env., 15.XII. 2013, 1 ex.; Vilaflor: 14.XII.2018, 1 ex.; Area Recreativa Chio: 09.IX.2019, 9 exx. (8 adults, 1 larva) collected from *Pinus canariensis;* San Miguel de Abona: 2.II.2020, 2 exx.; Santa Cruz De Tenerife: 23.II.2022, 4 exx. (3 adults, 1 larva); Montaña de Joco: 1.II.2023, 3 exx.; Mirador Piedra la Rosa: 1.II.2023, 1 ex.

**La Gomera.** San Sebastian: 1.III.1997, 11 exx. (leg. J. Borowski); Hermigua: 3.III.1997, 2 exx. (leg. J. Borowski); Las Hayas: 6.II.2020, 3 exx.; Mirador de Tajaqué: 4.II.2020, 1 ex.; Zona Recreativa Laguna Grande 6.II.2020, 1 ex.; San Sebastián de la Gomera: 7.II.2020, 1 ex.; Alajeró, Casas Rurales Los Manantiales, Mirador de Igualero, Playa Santiago, La Dama, San Sebastián de La Gomera, Valle Gran Rey: 24.II–28.II.2022, a total of 84 exx. (75 adults, 9 larvae) collected from *Pinus canariensis,* herbaceous plants and unidentified Fabaceae.


***Coccinella septempunctata algerica* Kovář, 1977**


**Tenerife.** Montaña de Joco: 1.II.2023, 2 exx.

**La Gomera.** Las Hayas: 12.I.2020, 1 ex.; Alto de Garajonay: 13.I.2020, 1 ex.; Raso de la Bruma: 3.II.2020, 1 ex.; San Sebastián de La Gomera: 24.II.2022, 3 exx. from *Tamarix* sp. and *N. oleander*; Alajeró: 25.II.2022, 1 ex. (pupa).


***Harmonia axyridis* (Pallas, 1773)**


**Tenerife.** Puerto de la Cruz, Botanical Garden: 30.XI.2021, 1 ex.; Puerto de la Cruz, Parque Taoro: 20.XI.2021, 1 ex.; Santa Cruz De Tenerife: 23.II.2022, 28 exx. (9 adults, 19 larvae) collected from *Ficus* sp., *Schefflera* sp. and *Ricinus communis.*

**Remark.** A well-known invasive species of Asiatic origin, currently nearly cosmopolitan [21].


***Hippodamia variegata* (Goeze, 1777)**


**Tenerife.** La Caldera, La Esperanza, Los Pinos, Santiago del Teide, Vilaflor: 08.IX–11.IX.2019, 28 exx. (24 adults, 4 larvae) collected from *Hibiscus* sp., *N. oleander* and *Citrus* sp.; Santa Cruz De Tenerife: 14.XII.2018, 1 ex., 23.II.2022, 3 exx. collected from *N. oleander*; Los Christianos: 6.VI.2022, 1 ex. collected from *Coccoloba uvifera* (leg. A. Krzysztofiak).

**La Gomera.** San Sebastian: 1.III.1997, 7 exx. (leg. J. Borowski); Agulo: 27.II.2022, 1 ex.; Valle Gran Rey: 26.II.2022, 1 adult and 6 larvae collected from *Hibiscus* sp. and *N. oleander*.


***Myrrha octodecimguttata* (Linnaeus, 1758)**


**Tenerife.** Las Vegas: 10.IX.2019, 3 exx. collected from *Pinus canariensis*; Arico: 31.I.2020, 2 exx.; Mirador Piedra la Rosa: 1.II.2023, 1 ex.

**Remark.** It has not previously been reported to occur on Tenerife.


***Oenopia doublieri* (Mulsant, 1846)**


**La Gomera.** Agulo: 27.II.2022, 1 ex. collected from *Eriobotrya japonica* (Thunb.) Lindl.

**Remark.** First report of this species from La Gomera.


***Olla v-nigrum* (Mulsant, 1866)**


**Tenerife.** La Orotava: 24.III.2011, 1 ex. (leg. J.M. Vela); Aeropuerto de Tenerife Sur: 22.VI.2021, 2 exx.

**La Gomera.** San Sebastián de La Gomera: 7.II.2020, 5 exx., 25.II.2022, a total of 99 exx. (38 adults, 21 pupae, 40 larvae) collected from *Tamarix* sp., *Cyperus* sp. and herbaceous plants; Playa Santiago: 24.II.2022, 20 pupae and 27 larvae collected from woody Fabaceae, *Tamarix* sp. and *Olea europaea* L.

**Remark.** A species native to North, Central and South America [22], here reported for the first time from La Gomera.

Noviini Mulsant, 1846


***Novius cardinalis* (Mulsant, 1850)**


**Tenerife.** Arico, El Sauzal, San Cristóbal de La Laguna, Puerto de la Cruz, Vilaflor: 8.IX–11.IX.2019, 8 exx. collected from *N. oleander* and *Hibiscus* sp.; Santa Cruz De Tenerife: 22.VI.2021, 22 exx. collected from *Bambusa* sp., palm trees and *Ficus* sp.; Aeropuerto de Tenerife Sur: 22.VI.2021, 3 exx.; Los Roques: 31.I.2023, 1 ex.; Las Maretas: 6.II.2023, 1 ex.; Los Silos: 4.II.2023, 1 ex.; El Chorrillo: 30.I.2023, 1 ex.

**La Gomera.** El Cedro: 7.II.2020, 1 ex.; La Gerode: 5.II.2020, 1 ex.; Agulo, Hermigua, La Dama, Playa Santiago, San Sebastián de La Gomera, Valle Gran Rey: 24.II–28.II.2022, a total of 24 exx. (17 adults, 7 larvae) collected from *Tamarix* sp., *Cyperus* sp. and herbaceous plants.

**Remark.** A species native to Australia, introduced as a biocontrol agent in many warmer regions throughout the world [23].


***Novius cruentatus* (Mulsant, 1846)**


**Tenerife.** Area Recreativa Las Lajas: 9.IX.2019, 2 exx.; Las Vegas: 10.IX.2019, 22 exx. collected from *Pinus canariensis;* Vilaflor: 21.XI.2021, 8 exx. collected from *P. canariensis*.

Scymnini Mulsant, 1846


***Clitostethus arcuatus* (P. Rossi, 1794)**


**Tenerife.** Santa Cruz De Tenerife: 23.II.2022, 37 exx. collected mostly from *Ricinus communis*.


***Nephaspis bicolor* Gordon, 1982**


**Tenerife.** Santa Cruz De Tenerife: 23.II.2022, 7 exx. collected from *Bambusa* sp., palm trees and *Ficus* sp.; Las Maretas: 6.II.2023, 25 exx. collected from *Ficus* sp. and palm tree (feeding on *Aleurodicus* sp.); Puerto de la Cruz, Parque Taoro: 24.XI.2023, 10 exx. (feeding on *Aleurodicus* sp.).

**Remark.** A species native to Trinidad, introduced into Hawaii [24] and the Canary Islands [25].


***Nephus* (*Geminosipho*) *reunioni* (Fürsch, 1974)**


**Tenerife.** Arico, Arona, El Sauzal, San Cristóbal de La Laguna, Tegueste, Vilaflor: 9.IX–11.X.2019, 37 exx.; Arico: 31.I.2020, 1 ex.; Aeropuerto de Tenerife Sur: 22.VI.2021, 2 exx.; Puerto de la Cruz, Botanic Garden: 30.XI.2021, 1 ex.; Santa Cruz De Tenerife: 23.II.2022, 9 exx. collected from palm trees; Las Maretas, 6.II.2023, 16 exx.; Puerto de la Cruz: 24.XI.2023, 5 exx.

**Remark.** This widely used biocontrol agent is endemic to the island of Réunion (Mascarene Islands) [26]. Within the Canary Islands, it has previously been reported only from La Palma [15].


***Nephus* (*Nephus*) *flavopictus* (Wollaston, 1854)**


**Tenerife.** Barranco de la Torre: 12.XII.2016, 1 ex.; Las Lagunetas: 13.XII. 2016, 1 ex.; Aeropuerto de Tenerife Sur, Arico, El Sauzal, La Cardera, Puerto de la Cruz, San Cristóbal de La Laguna, Santiago del Teide, Tamaimo: 8.IX–11.IX.2019, 20 exx. collected from *N. oleander, Euphorbia* sp., *Ficus sp.* and *Launaea arborescens* (Batt.) Murb.; Granadilla: 2.II.2020, 4 exx.; El Chorrillo: 1.II.2020, 2 exx.; Fasnia: 31.I.2020, 1 ex.; Arico: 31.I.2020, 1 ex.; Aeropuerto de Tenerife Norte: 15.IV.2021, 1 ex. collected from *C. equisetifolia*; Aeropuerto de Tenerife Sur: 22.VI.2021 and 1.III.2022, total of 21 exx.; Santa Cruz De Tenerife: 23.II.2022, 3 exx.; Las Maretas: 6.II.2023, 11 exx., 28.XI.2023, 2 exx.; Playa del Socorro: 26.XI.2023, 2 exx.

**La Gomera.** Alto de Garajonay: 13.I.2020, 1 ex.; La Gerode: 19.I.2020, 18 exx.; Hermigua: 3.II.2020, 3 exx.; El Molinito, San Sebastián de La Gomera, Vallehermoso: 24.II–27.II.2022, a total of 11 exx. collected from *Tamarix* sp., *N. oleander* and succulent plants; Hermingua: 20.III. 2022, 2 exx.


***Nephus* (*Nephus*) *incisus* (Har. Lindberg, 1950)**


**Tenerife.** Area Recreativa Chio, Arico, La Esperanza, Las Galletas: 9.IX–11.X.2019, 6 exx. collected mostly from *N. oleander*, *Hibiscus* sp. and *Ricinus communis*; Santa Cruz De Tenerife: 23.II.2022, 1 ex. collected from *Ricinus communis*; Las Maretas: 6.II.2023, 7 exx.

**La Gomera.** San Sebastián de La Gomera, Playa Santiago, Santa Ana, Valle Gran Rey, Vallehermoso: 24.II–27.II.2022, a total of 71 exx. collected mostly from *N. oleander* and *Olea europaea* L.


***Scymnus* (*Mimopullus*) *cercyonides* Wollaston, 1864**


**Tenerife.** Tabaiba: 12.XII.2016, 1 ex. collected from *Erica* sp.; Roque del Conde: 14.XII.2016, 1 ex.; Masca: 17.XII. 2016, 4 exx.; Montana de Tafada: 27.XII.2017, 4 exx.; Icor: 12.X.2018, 6 exx.; Chamorga: 15.XII. 2018, 1 ex.; Puerto de la Cruz: 8.IX.2019, 1 ex.; Granadilla: 8.IX.2019, 8 exx. collected from *N. oleander* and unidentified ornamental plants; Vilaflor: 21.XI.2021, 3 exx.; Buenavista del Norte: 23.XI.2021, 2 exx.; Las Maretas: 6.II.2023, 1 ex.

**La Gomera.** Teselinde: 17.I.2020, 1 ex.; Agulo: 5.II.2020, 1 ex.


***Scymnus* (*Mimopullus*) *marinus* Mulsant, 1850**


**Tenerife.** El Bailadero: 17.XII.2013, 1 ex. (det. I. Kovář); Barranco de Ajeque: 12.XII.2016, 1 ex. (det. I. Kovář).


***Scymnus* (*Pullus*) *canariensis* Wollaston, 1864**


**Tenerife.** Puertito de Guimar: 17.XII. 2018, 2 exx.; Aeropuerto de Tenerife Sur, Arico, Arona, Chimiche, Chirche, El Sauzal, Granadilla, La Esperanza, La Cardera, La Sabinita, Las Galletas, Las Vegas, Los Pinos, Puerto de la Cruz, Puerto de Santiago, Santiago del Teide, San Cristóbal de La Laguna, Tegueste, Tamaimo, Vilaflor: 8.IX–11.IX.2019, total of 502 exx. collected mostly from *Pinus canariensis*, *N. oleander, Hibiscus* sp. and *Ficus* sp.; Aeropuerto de Tenerife Norte: 15.VI.2021, 1 ex.; Aeropuerto de Tenerife Sur: 22.VI.2021, 7 exx.; Santa Cruz De Tenerife: 23.II.2022, 5 exx.; Los Roques: 31.I.2023, 1 ex.; Las Maretas: 6.II.2023, 1 ex.

**La Gomera.** Tagamiche mt.: 16.I.2020, 2 exx.; La Gerode: 19.I.2020, 3 exx.; Aeropuerto de La Gomera, Agulo, Alojera, Mirador de El Palmarejo, Playa Santiago, La Dama, San Sebastián de La Gomera, Valle Gran Rey: 24.II–1.III.2022, total of 81 exx (80 adults, 1 larva) collected from various plants including *Rumex lunaria*, *N. oleander, Pinus canariensis*, *Euphorbia* sp., *Arundo donax* L. and herbaceous vegetation; Hermigua: 20.III. 2022, 11 exx.


***Scymnus* (*Pullus*) *medanensis* Eizaguirre, 2007**


**Tenerife.** Aeropuerto de Tenerife Sur: 10.IX.2019, 4 exx. collected from *Rumex lunaria* and *Euphorbia* sp.


***Scymnus* (*Scymnus*) *nubilus* Mulsant, 1850**


**Tenerife.** Barranco de Acetenjo: 25.XII.2017, 5 exx.; Arico, Arona, El Sauzal, Las Galletas, Los Pinos, Puerto de Santiago, Tegueste: 8.IX–11.IX.2019, total of 21 exx. collected mostly from *N. oleander*, *Ficus* sp., *Hibiscus* sp. and *A. donax;* Santa Cruz De Tenerife: 23.II.2022, 1 ex.; Los Silos: 18.XI.2021, 1 ex.; Puerto de la Cruz: 20.XI.2021, 2 exx.

**La Gomera.** Hermigua: 3.II.2020, 3 exx.; Agulo, Playa Santiago, Valle Gran Rey: 25.II–27.II.2022, 6 exx.

Hyperaspidini Mulsant, 1846


***Hyperaspis vinciguerrae* Capra, 1929**


**Tenerife.** Aeropuerto de Tenerife Sur: 22.VI.2021, 2 exx. collected from *Bougainvillea* sp.

**Remark.** This species has so far been reported from the three easternmost islands of the Canary archipelago: Fuerteventura, Lanzarote and Gran Canaria. Tenerife is the next island with its documented presence. *Hyperaspis vinciguerrae* is probably alien to the Canary Islands, occurring natively in NE Africa and the Middle East [11].

Diomini Gordon, 1999

***Diomus gillerforsi* Fürsch, 1987** (Figure 2A and Figure 3)

**Tenerife.** Barranco de Ajeque: 12.XII.2016, 1 ex. (male) collected from a succulent.

***Diomus rubidus* Motschulsky, 1837** (Figure 2B and Figure 4)

**Tenerife.** El Guincho: 31.VII. 2005, 5 exx. (2 males and 3 females) collected from xerotherm shrubs.

Azyini Mulsant, 1850


***Cryptolaemus montrouzieri* Mulsant, 1853**


**Tenerife.** Puerto de la Cruz: 15.III.2008, 1 ex., 30.XI.2021, 1 ex.; Afur env.: 4.II.2017, 1 ex., 30.XI.2021, 1 ex.; Las Maretas: 6.II.2023, 6 exx.

**La Gomera.** Agulo, Hermigua, San Sebastián de La Gomera, Santa Ana, Valle Gran Rey, Vallehermoso: 24.II–27.II.2022, total of 40 exx (38 adults, 2 larvae) collected from various plants, including *Tamarix* sp., *N. oleander* and *Opuntia ficus-indicia*.

**Remarks.** A species native to Australia that has been widely used as a biocontrol agent against mealybugs and has consequently spread throughout warmer regions of the world [27].

Chilocorini Mulsant, 1846


***Chilocorus canariensis* Crotch, 1874**


**Tenerife.** Degollada de Cherfe: 13.XII.2016, 1 ex.; Tamaimo: 10.XII.2016, 3 exx.; Arona, Chirche, Tamaimo: 9.IX–10.IX.2019, 15 exx. collected mostly from *Euphorbia* sp.; Santa Cruz De Tenerife: 23.II.2022, 2 larvae from *N. oleander*; Aeropuerto de Tenerife Sur: 1.III.2022, 1 ex. from *Schefflera* sp.; Adeje: 4.II.2023, 3 exx.; Arico: 31.I.2020, 4 exx.; El Chorrillo: 1.II.2020, 1 ex.; Las Eras: 26.XI.2021, 2 exx.; Buenavista del Norte: 2.II.2023, 2 exx.

**La Gomera.** San Sebastian: 1.III.1997, 1 ex. (leg. J. Borowski); Hermigua: 3.III.1997, 3 exx. (leg. J. Borowski); La Gerode: 5.II.2020, 1 ex.; Agulo: 6.II.2020, 1 ex.; El Palmar, 5.II.2020, 1 ex.; Hermigua: 20.III. 2022, 3 exx.


***Parexochomus nigripennis* (Erichson, 1843)**


**Tenerife.** Arona, El Sauzal, Las Galletas, Santiago del Teide, Tegueste: 8.IX–11.IX.2019, total of 27 exx. collected mostly from *Opuntia ficus-indica* (L.) Mill., *P. dulcus*, *Bougainvillea* sp. and *N. oleander*; El Chorrillo: 30.I.2023, 2 exx.

**La Gomera.** Aeropuerto de La Gomera, Agulo, Playa Santiago, San Sebastián de La Gomera, Santa Ana, Valle Gran Rey: 23.II–27.II.2022, total of 8 exx. were collected from *Tamarix* sp., *Cyperus* sp. and other plants; Hermigua: 20.III. 2022, 1 ex.

Sticholotidini Pope, 1962

***Coelopterus* sp.** (Figure 2C,D)

Tenerife. Cruz del Carmen: 24.XI.2021, 1 female collected from *Laurus* sp. in laurisilva.

**Remark.** First report of the genus *Coelopterus* in the Canary Islands.


***Pharoscymnus decemplagiatus* (Wollaston, 1857)**


**Tenerife.** Arico, Arona, Chirche, El Sauzal, La Cardera, La Esperanza, Las Galletas, Las Vegas, Los Pinos, Puerto de la Cruz, Puerto de Santiago, Tamaimo, Vilaflor: 8.IX–11.IX.2019, total of 114 exx. collected mostly from *P. canariensis*, palm trees and *N. oleander*; Arico: 31.I.2020, 3 exx.; El Chorrillo: 1.II.2020, 3 exx.; Granadilla: 2.II.2020, 1 ex.; Aeropuerto de Tenerife Norte: 15.VI.2021, 3 exx. from *Casuarina equisetifolia* L.; Vilaflor: 21.XI.2021, 1 ex.; Buenavista del Norte: 23.XI.2021, 3 exx., 3.II.2023, 1 ex.; Arico: 22.II.2022, 4 exx. collected from *Pinus canariensis*; Las Maretas: 6.II.2023, 2 exx.

**La Gomera.** Barranco de Argaga: 10.I.2020, 2 exx.; La Gerode: 19.I.2020, 2 exx.; El Monlinito, Playa Santiago, San Sebastián de La Gomera, Valle Gran Rey: 24.II–26.II.2022, 7 exx. collected mostly from *Cedrus* sp.; Hermigua: 20.III. 2022, 1 ex.


***Pharoscymnus flexibilis* Mulsant, 1853**


**Tenerife.** Los Batanes env.: 31.I.2023, 1 ex. collected from *Rubus* sp.

**Remark.** A species distributed in southern and western Asia, recently reported from two islands of the Canary archipelago, Fuerteventura [11] and Lanzarote [12]. It has not previously been reported from Tenerife.

Coccidulini Mulsant, 1846


***Rhyzobius litura* (Fabricius, 1787)**


**Tenerife.** Buenavista del Norte: 3.II.2023, 1 ex.; Las Portelas: 3.II.2023, 2 exx.

**La Gomera.** Epina: 17.I.2020, 1 ex., 4.II.2020, 1 ex.; La Gerode: 5.II.2020, 1 ex.; Valle Gran Rey: 4.II.2020, 1 ex.; Las Hayas: 6.II.2020, 1 ex.; Casas Rurales Los Manantiales: 25.II.2022, 1 ex.


***Rhyzobius lophanthae* (Blaisdell, 1892)**


**Tenerife.** Las Maretas: 6.II.2023, 1 ex.; Afur env.: 4.II.2017, 1 ex.; Area Recreativa Chio, Arico, Arona, El Sauzal, La Esperanza, La Cardera, Las Galletas, Los Pinos, Puerto de la Cruz, San Cristóbal de La Laguna, Tamaimo, Tegueste: 8.IX–11.IX.2019, total of 68 exx. (66 adults, 2 larvae) collected mostly from *Cycas* sp., palms and *N. oleander*; Santa Cruz De Tenerife: 23.II.2022, 6 exx. collected from palms, *N. oleander* and *Ricinus communis*.

**La Gomera.** Playa Santiago, San Sebastián de La Gomera, Santa Ana, Valle Gran Rey, Vallehermoso: 24.II–27.II.2022, total of 35 exx. collected mostly from *N. oleander*, *Cycas* sp., *Ricinus communis* and *O. europaea*.

**Remark.** A species alien to the Canary Islands. It is native to Australia but was introduced worldwide for the biological control of scale insects [28].

Tetrabrachyni Kapur, 1948


***Tetrabrachys deserticola* (Wollaston, 1864)**


**Tenerife.** Las Maretas: 6.II.2023, 1 ex.; Punta del Sol: 28.I.2010, 1 ex. (leg. A. Machado); Las Maretas: 3.II.2020, 5 exx., 21.XI.2021, 1 ex.

Comparing the number of individuals of species observed during this survey on both islands, it can be seen that Macaronesian endemics and species alien to the region were the most numerous (Figure 5). On Tenerife, the endemic *Scymnus canariensis* clearly predominated, followed by the alien *Delphastus catalinae* and the endemic *Pharoscymnus decemplagiatus*. The two species most frequently recorded on La Gomera (*Olla v-nigrum* and *D. catalinae*) are alien, and the next two (*Coccinella miranda* and *S. canariensis*) are endemic to the Canary Islands.

## 4. Discussion

Our study documents the occurrence of markedly more ladybird species on the larger island (Tenerife, 35 species) that on the smaller one (La Gomera, 20 species). Five of the species (*Myrrha octodecimguttata*, *Nephus reunioni*, *Pharoscymnus flexibilis*, *Coelopterus* sp. and *Hyperaspis vinciguerrae*) are reported for the first time from Tenerife and two (*Oenopia doublieri* and *Olla v-nigrum*) from La Gomera (Table 1). The record of the genus *Coelopterus* is the first for the whole Canary archipelago. The single female collected could not be identified with certainty to the species level, but based on the reported distribution of the three Palaearctic species of *Coelopterus*—*C. armeniacus* Weise (Armenia, Israel), *C. desertorum* Dobzhansky (Kazakhstan) and *C. salinus* Mulsant & Rey (SW Europe, North Africa, Iran and the Afrotropical region) [30]—we can surmise that it probably belongs to the latter species.

In this study, we also provide the earliest published observations for the Canary Islands of two alien species, *O. v-nigrum* and *N. reunioni*. The former has been recorded in the archipelago since 2014 [13], but our data from Tenerife push this date back three years (March 2011). *Nephus reunioni*, recorded in several localities on La Palma in June 2021 [15], is reported here as being present on Tenerife already in September and October 2019. Another alien species, the harlequin ladybird (*H. axyridis*), highly invasive in many parts of the world [21], was recorded on Tenerife in 2003 and 2004 [10,31], and its next records known to us are those from 2021 and 2022 reported here. Interestingly, while on Tenerife, *H. axyridis* appears to be well established (a reproducing population recorded) and associated with its specific natural enemies [32,33], no other island of the Canarian archipelago has been reported to host this ladybird. It can be assumed that the species will appear on other islands of the archipelago in the near future.

Some ladybirds previously reported as occurring on Tenerife (12 species) and/or La Gomera (6 species) were not recorded in this study. The inclusion of these species brings the total number of Coccinellidae reported from Tenerife to 47 and from La Gomera to 26 (Table 1). However, the current occurrence in the Canary Islands of certain species in this group needs to be confirmed by up-to-date records. For example, *Adalia testudinea* (Wollaston, 1854), a species described and mainly reported from Madeira [34,35,36], was also mentioned as occurring in the Canary Islands, including Tenerife. A single specimen was reported from this island by Iablokoff-Khnzorian [37] and another from Gran Canaria by Eizaguirre [10]. *Vibidia duodecimguttata* (Poda, 1761) has also been reported from the archipelago very sporadically: a single specimen was collected a century ago in Gran Canaria [5] and was reported from Tenerife without any details in the checklist by Oromí et al. [2]. Other species of uncertain status are *Nephus binotatus* (Brisout, 1863) (two specimens collected on Tenerife in 1995 [10]), *N. conjunctus* (Wollaston, 1870) (single reports from Tenerife and Gran Canaria [9]), *Nephus depressiusculus* (Wollaston, 1867) (reports from Gran Canaria [9,10]; Tenerife [10]; and with no details (a checklist), from La Palma [2]) and *Scymnus interruptus* (Goeze, 1777) (reports from Tenerife [9,10]). None of the species mentioned above were found during our recent surveys on all seven main islands of the archipelago. Without new data confirming their presence in Tenerife and/or other Canarian islands, we cannot conclude that they are established there.

**Table 1 insects-15-00596-t001:** Coccinellidae recorded on Tenerife and La Gomera. Only the first reports on each species are quoted in the ‘literature data’ columns. Species new to either Tenerife or La Gomera are in bold print.

	Tenerife		La Gomera	
Species	This Study	Literature Data	This Study	Literature Data
*Delphastus catalinae* (Horn)	**+**	[10]	**+**	[10]
*Stethorus tenerifensis* Fürsch	**+**	[9]	**+**	[2]
*Stethorus wollastoni* Kapur	**+**	[3]	**+**	[3]
*Adalia bipunctata* (L.)		[38]		[10]
*Adalia decempunctata* (L.)	**+**	[10]		
*Adalia testudinea* (Wollaston)		[37]		
*Cheilomenes propinqua* (Mulsant)	**+**	[10]		
*Coccinella miranda* Wollaston	**+**	[3]	**+**	[3]
*Coccinella septempunctata algerica* Kovář	**+**	[3]	**+**	[3]
*Harmonia axyridis* (Pallas)	**+**	[10]		
*Hippodamia variegata* (Goeze)	**+**	[7]	**+**	[2]
***Myrrha octodecimguttata* (L.)**	**+**			[39]
***Oenopia doublieri* (Mulsant)**		[40]	**+**	
***Olla v-nigrum* (Mulsant)**	**+**	[41]	**+**	
*Vibidia duodecimguttata* (Poda)		[2]		
*Novius cardinalis* (Mulsant)	**+**	[7]	**+**	[38]
*Novius cruentatus* (Mulsant)	**+**	[38]		
*Novius canariensis* Korschefsky		[38]		
*Clitostethus arcuatus* (Rossi)	**+**	[3]		[38]
*Nephaspis bicolor* Gordon	**+**	[25]		
*Nephus* (*Bipunctatus*) *conjunctus* (Wollaston)		[9]		
***Nephus* (*Geminosipho*) *reunioni* (Fürsch)**	**+**			
*Nephus* (*Nephus*) *binotatus* (Brisout)		[10]		
*Nephus* (*Nephus*) *flavopictus* (Wollaston)	**+**	[3]	**+**	[4]
*Nephus* (*Nephus*) *incisus* (Har. Lindberg)	**+**	[8]	**+**	[39]
*Nephus* (*Sidis*) *depressiusculus* (Wollaston)		[2]		
*Scymnus* (*Mimopullus*) *cercyonides* Wollaston	**+**	[3]	**+**	[3]
*Scymnus* (*Mimopullus*) *marinus* (Mulsant)	**+**	[9]		[9]
*Scymnus* (*Pullus*) *canariensis* Wollaston	**+**	[3]	**+**	[3]
*Scymnus* (*Pullus*) *medanensis* Eizaguirre	**+**	[10]		
*Scymnus* (*Pullus*) *subvillosus* (Goeze)		[9]		[9]
*Scymnus* (*Scymnus*) *interruptus* (Goeze)		[9]		
*Scymnus* (*Scymnus*) *nubilus* Mulsant	**+**	[9]	**+**	[9]
*Scymnus* (*Scymnus*) *rubromaculatus* (Goeze)		[39]		
*Scymnus* (*Scymnus*) *rufipennis* Wollaston		[3]		[3]
***Hyperaspis vinciguerrae* Capra**	**+**			
*Diomus gillerforsi* Fürsch	**+**	[9]		
*Diomus rubidus* Motschulsky	**+**	[42]		
*Cryptolaemus montrouzieri* Mulsant	**+**	[38]	**+**	[43]
*Chilocorus canariensis* Crotch	**+**	[3]	**+**	[3]
*Parexochomus nigripennis* (Erichson)	**+**	[38]	**+**	[38]
***Coelopterus* sp.**	**+**			
*Pharoscymnus decemplagiatus* (Wollaston)	**+**	[3]	**+**	[4]
***Pharoscymnus flexibilis* Mulsant**	**+**			
*Rhyzobius litura* (Fabricius)	**+**	[3]	**+**	[4]
*Rhyzobius lophanthae* (Blaisdell)	**+**	[44]	**+**	[10]
*Tetrabrachys deserticola* (Wollaston)	**+**	[38]		
No. species	35	42	20	24
Total no. species	47		26	

The documented species richness of Coccinellidae on Tenerife (47 species) is the highest of all the Canary Islands, followed by Gran Canaria, where 41 species have been recorded (42 species previously reported [13], but 1 of these, *Novius conicollis* Korschefsky, was synonymized with *N. cruentatus* (Mulsant) [15]). The lower number of ladybird species on La Gomera can be primarily attributed to its smaller size. In accordance with the theory of island biogeography [17], the relationship between ladybird species richness and island size appears to apply to the Canary Islands as a whole. We found a positive relationship between these variables (r = 8612, *p* = 0.03) (Figure 6), although the eastern islands, Fuerteventura and Lanzarote, have a clearly poorer ladybird fauna than would be expected from their size: only 27 species have been recorded on Fuerteventura, the second largest island in the archipelago, and 23 species on Lanzarote, the fourth largest island. Fuerteventura and Lanzarote are the oldest of the Canary Islands and lie closest to the African mainland [16], which should rather favour their species richness [17,45]. However, they are also more arid and less elevated than the remaining islands and have lower habitat diversity with the lack of more complex habitats, such as laurel or pine forests [16,18]. Possibly due to this low habitat diversity and these harsh conditions, immigrants from Africa and Europe establish on Fuerteventura and Lanzarote less frequently than on some other islands of the archipelago, such as Tenerife, Gran Canaria and La Palma.

Testing several models of biogeographical processes on empirical data of various Canarian plant and animal lineages, Sanmartín et al. [46] came to the conclusion that the central islands (especially Tenerife and Gran Canaria) act as ‘centres of diversification and dispersal’ within the archipelago and show the highest rate of exchange with the mainland. On the other hand, the eastern islands closest to the mainland (Fuerteventura and Lanzarote) have low carrying capacity and, despite their relatively large size and older age, lower species richness compared to the more western islands. Our results seem to be in line with these findings. The spread of *Pharoscymnus flexibilis* from the African mainland to Fuerteventura, Lanzarote and Tenerife ([11,12,47] this study) is perhaps an example of the classical stepping-stone dispersal model [17], but this kind of dispersal is rather uncommon among the Coccinellidae in the Canary Islands. The ongoing colonization of the archipelago by other species, such as *Olla v-nigrum*, *Cheilomenes propinqua* or *Harmonia axyridis*, indicates the importance of the central islands (Tenerife and Gran Canaria) to this process ([13] this study).

Of all the Canary Islands, Tenerife appears to be the island of greatest importance as a recipient of ladybird immigrants, including alien species from remote regions. So far, ten alien ladybird species have been recognized as established in the Canary Islands, and for almost all of them, Tenerife was the island of their first record within the archipelago (Table 2). All ten of these alien species have been reported to occur on Tenerife, seven each on Lanzarote and La Palma, six each on Fuerteventura and Gran Canaria, five on La Gomera and three on El Hierro. Tenerife is therefore not only the species-richest of the Canary Islands but also the most susceptible to colonization by non-native ladybirds.

**Table 2 insects-15-00596-t002:** First records and current distribution in the Canary Islands of alien species of Coccinellidae. Island symbols: L, Lanzarote; F, Fuerteventura; GC, Gran Canaria; T, Tenerife; G, La Gomera; P, La Palma; H. El Hierro. The distribution data were taken from our recent papers [11,12,13,14,15] and this study.

	First Record in the Canary Islands			Current Distribution						
Species	Year	Island	Reference	L	F	GC	T	G	P	H
*Delphastus catalinae* (Horn)	before 2002	T, G	[48]	+	+	+	+	+	+	
*Harmonia axyridis* (Pallas)	2003	T	[10]				+			
*Olla v-nigrum* (Mulsant)	2011	T	this study	+		+	+	+	+	
*Novius cardinalis* (Mulsant)	1931	T	[7]	+	+	+	+	+	+	+
*Nephaspis bicolor* Gordon	2015	T	[25]				+		+	
*Nephus reunioni* (Fürsch)	2019	T	this study				+		+	
*Cryptolaemus montrouzieri* Mulsant	1966	T	[38]	+	+	+	+	+	+	+
*Pharoscymnus flexibilis* Mulsant	2016	F	[47]	+	+		+			
*Rhyzobius lophanthae* (Blaisdell)	1964	T	[44]	+	+	+	+	+	+	+
*Hyperaspis vinciguerrae* Capra	1925–1927	GC	[5]	+	+	+	+			

As for the number of endemic Macaronesian species, no such clear prevalence of Tenerife is marked. Of the 16 ladybird species recognized as endemic or near-endemic to Macaronesia (including *Parexochomus quadriplagiatus* and *Tetrabrachys deserticola*, which have also been reported from Morocco in NW Africa [30]), 14 have been recorded on Tenerife, slightly fewer than on Gran Canaria, where 15 species have been recorded (Table 3).

To conclude, our survey showed that the large island (Tenerife) hosts almost twice as many ladybird species as the small island (La Gomera) (47/26 = 1.81), and this approximate proportion also applies to endemic (14/8 = 1.75) and alien (10/5 = 2) species. The species–area relationship calculated for the seven main islands of the archipelago does give a significant correlation, but the data do not fit the model perfectly. In particular, the species richness in the second largest island, Fuerteventura, is clearly lower than expected, lying outside the area defined by the 95% confidence intervals. The markedly higher ladybird species richness on Tenerife than on the other Canarian islands is certainly a function of its large size, but other factors, such as habitat diversity and altitude variation, also play a part. The importance of Tenerife as a major recipient of alien species in the Canary Islands is likely to be reinforced by intensive tourism and trade-related transport.

## Figures and Tables

**Figure 1 insects-15-00596-f001:**
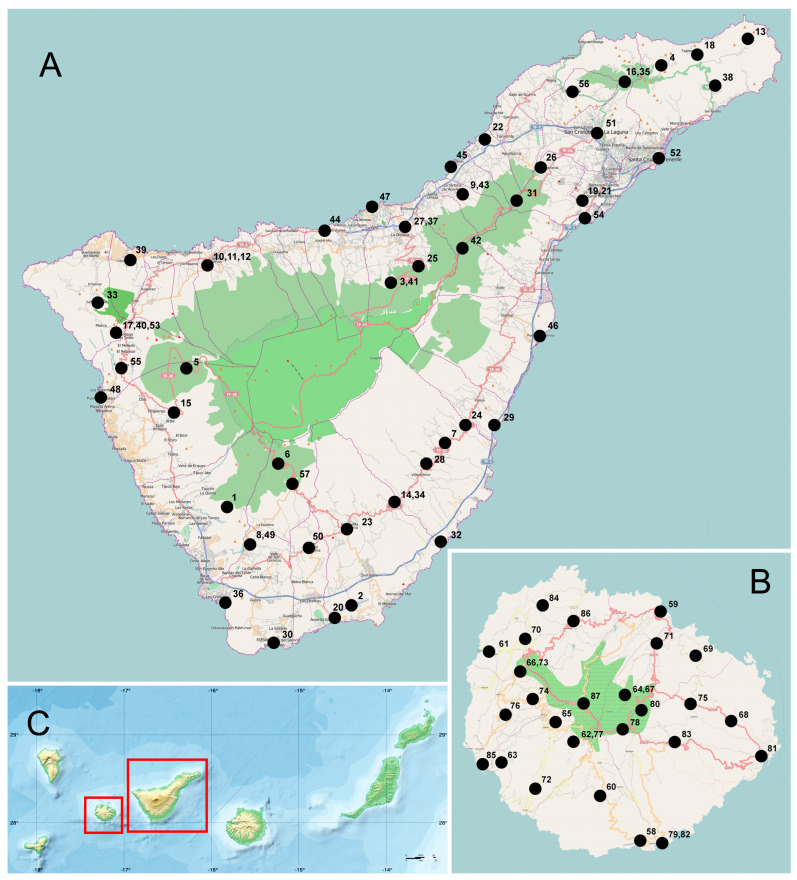
Collection sites of ladybird beetles on Tenerife (**A**) and La Gomera (**B**). (**C**) Location of both islands in the Canary Islands archipelago.

**Figure 2 insects-15-00596-f002:**
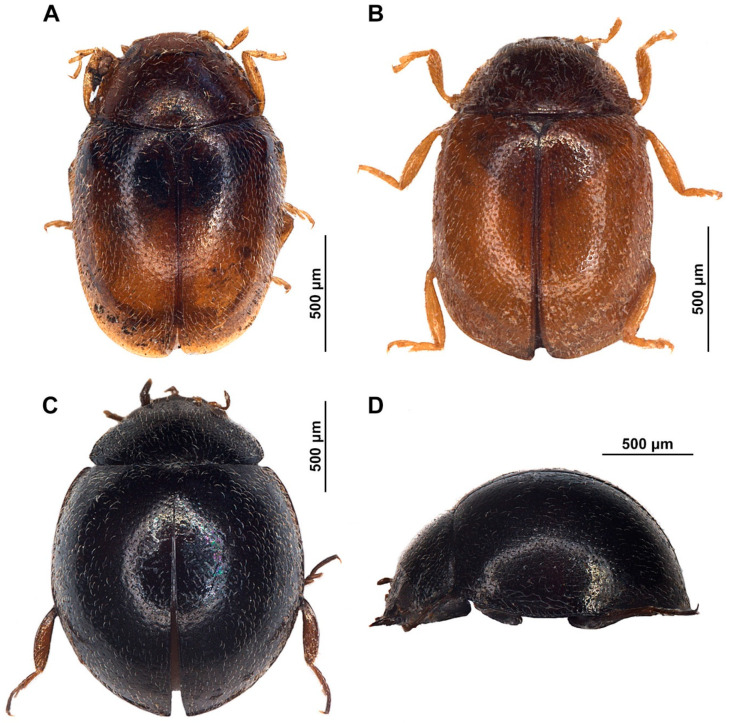
Habitus illustrations of (**A**) *Diomus gillerforsi* Fürsch; (**B**) *Diomus rubidus* (Motschulsky); (**C**) *Coelopterus* sp. female, dorsal; (**D**) *Coelopterus* sp. female, lateral.

**Figure 3 insects-15-00596-f003:**
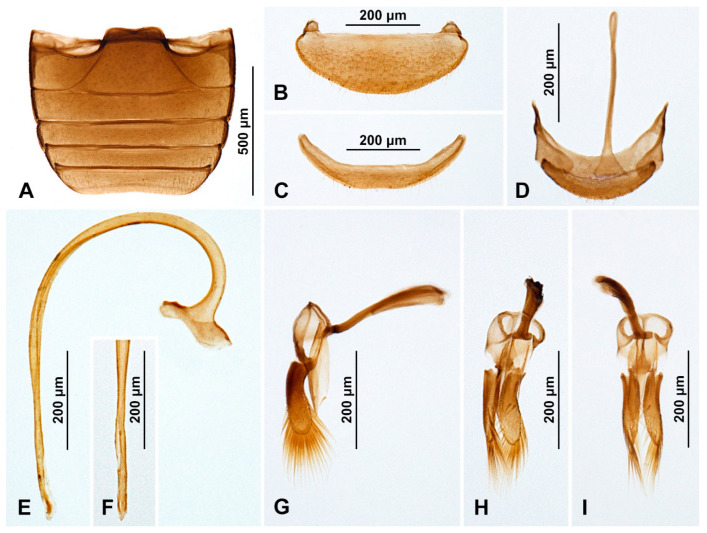
*Diomus gillerforsi* Fürsch, 1987, male. (**A**) Abdomen; (**B**) abdominal tergite VIII; (**C**) ventrite 6; (**D**) abdominal segments IX and X; (**E**) penis, lateral; (**F**) penis tip, inner; (**G**) tegmen, lateral; (**H**) tegmen, oblique view; (**I**) tegmen, inner.

**Figure 4 insects-15-00596-f004:**
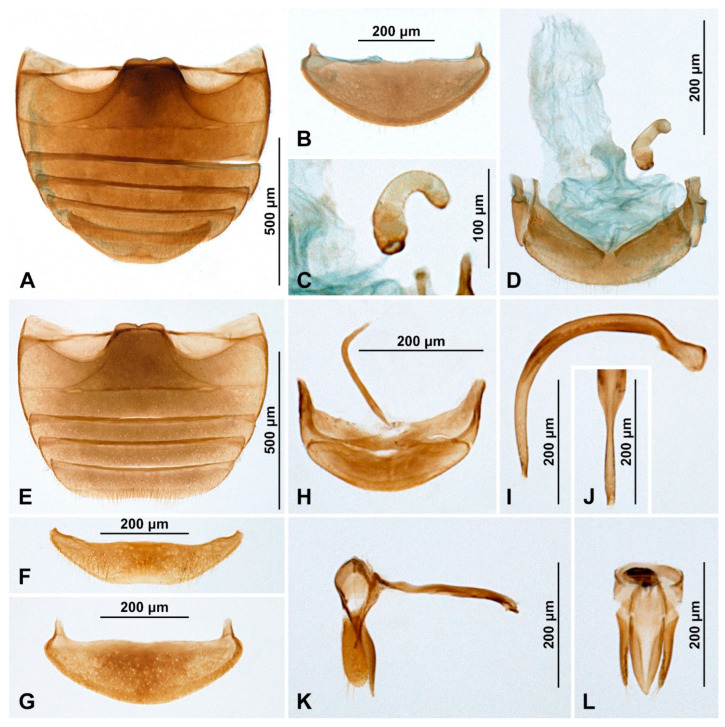
*Diomus rubidus* (Motschulsky, 1837). (**A**) Abdomen, female; (**B**) abdominal tergite VIII, female; (**C**) spermatheca; (**D**) female terminalia and genitalia; (**E**) abdomen, male; (**F**) ventrite 6, male; (**G**) abdominal tergite VIII, male; (**H**) abdominal segments IX and X; (**I**) penis, lateral; (**J**) penis tip, inner; (**K**) tegmen, lateral; (**L**) tegmen, inner.

**Figure 5 insects-15-00596-f005:**
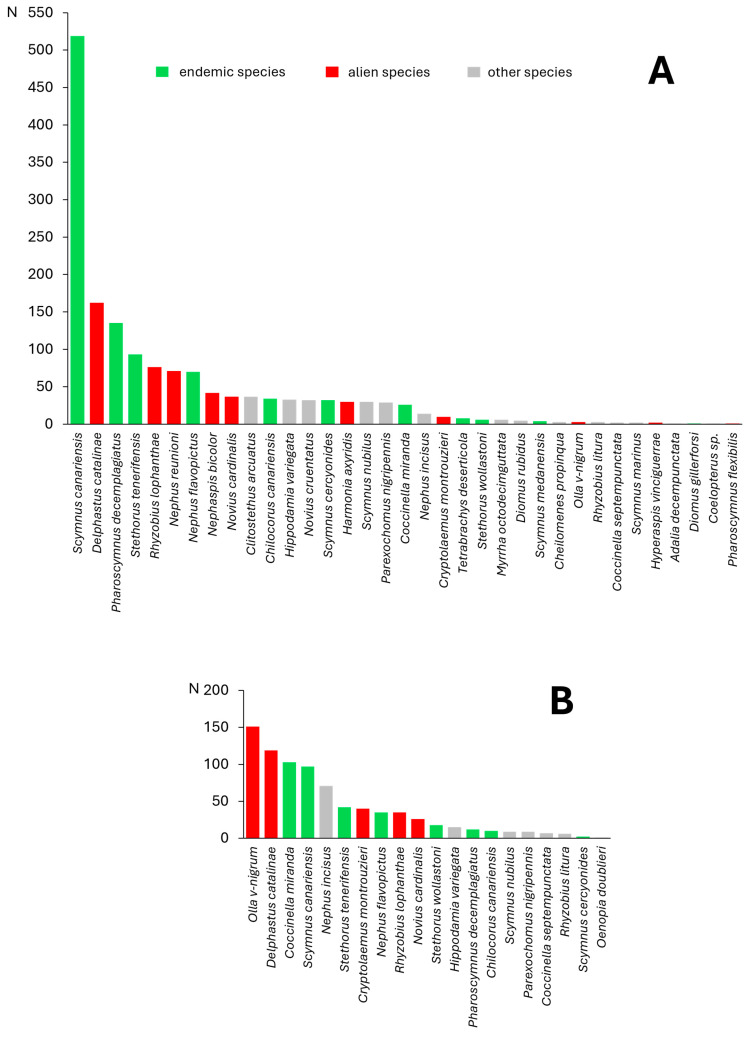
Species composition of Coccinellidae recorded in our survey on Tenerife (**A**) and La Gomera (**B**). The assignment of individual species to the distinguished zoogeographical categories (endemic species, alien species) was based on their distribution data provided or cited in our recent papers [11,12,13,14,15,29].

**Figure 6 insects-15-00596-f006:**
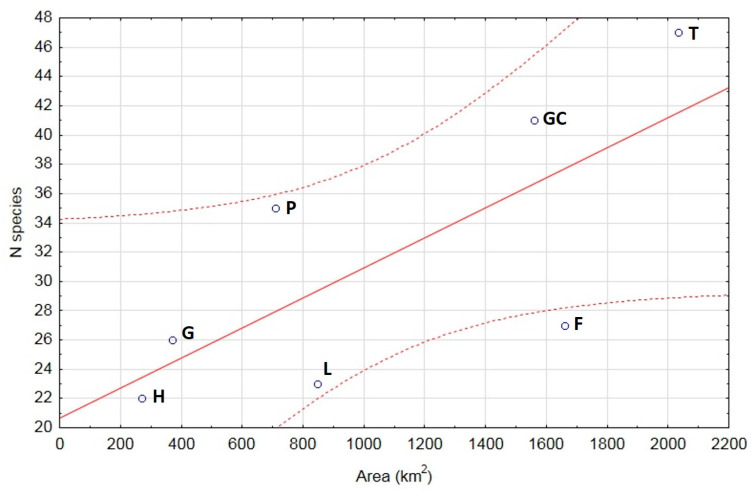
Linear regression with 95% confidence intervals (dotted lines) of the number of ladybird species in relation to the island area in the Canary archipelago. F = Fuerteventura, G = La Gomera, GC = Gran Canaria, H = El Hierro, L = Lanzarote, P = La Palma, T = Tenerife.

**Table 3 insects-15-00596-t003:** Species of Coccinellidae recorded on individual islands of the Canary Islands that are considered endemic or near-endemic to Macaronesia. Two species previously thought to be endemic to the Canary Islands [2,13] were not included: *Nephus incisus* (Lindberg), which appeared to have a much wider geographical range [49], and *Novius conicollis* Korschefsky, which was synonymized with *N. cruentatus* (Mulsant) [15]. Data on the distribution of *Diomus gillerforsi* came from the papers by Fürsch [9] and Eizaguirre [10], and this study. The distribution data for the remaining species were taken from our recent papers [11,12,13,14,15]. Island symbols as in Table 2.

Species	L	F	GC	T	G	P	H
*Stethorus tenerifensis* Fürsch	+	+	+	+	+	+	+
*Stethorus wollastoni* Kapur	+	+	+	+	+	+	+
*Adalia testudinea* (Wollaston)			+	+			
*Coccinella miranda* Wollaston		+	+	+	+	+	+
*Novius canariensis* Korschefsky			+	+		+	+
*Nephus depressiusculus* (Wollaston)			+	+		+	
*Nephus flavopictus* (Wollaston)	+	+	+	+	+	+	+
*Scymnus cercyonides* Wollaston			+	+	+	+	+
*Scymnus canariensis* Wollaston	+	+	+	+	+	+	+
*Scymnus medanensis* Eizaguirre	+	+	+	+			
*Diomus gillerforsi* Fürsch				+		+	
*Chilocorus canariensis* Crotch		+	+	+	+	+	+
*Parexochomus bellus* (Wollaston)			+				
*Parexochomus quadriplagiatus* (Wollaston)	+	+	+				
*Pharoscymnus decemplagiatus* (Wollaston)	+	+	+	+	+	+	+
*Tetrabrachys deserticola* (Wollaston)		+	+	+			
Number of species	7	10	15	14	8	11	9

## Data Availability

The data are contained within this article.

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
