# Peer review of "Diversity of Ladybird Beetles (Coleoptera: Coccinellidae) in Tenerife and La Gomera (Canary Islands): The Role of Size and Other Island Characteristics"

_insects, 2024, doi:10.3390/insects15080596_

Round 1
Reviewer 1 Report
Comments and Suggestions for Authors
- The manuscript is interesting, as is the study question... however the experimental design is deficient and poorly planned.
- Sampling was carried out in an irregular, non-systematized manner, which is why the data from both islands cannot be compared with each other.
- Nor is an estimate of collecting efforts presented, nor a species accumulation curve to estimate how many species could inhabit each island.
- With the methodology they followed, at most an updated faunal checklist can be presented.
- All results and their graphs should be placed in the corresponding section, not until the Discussion.
- Some species are mentioned as being of "special interest", but they do not explain why?
- They mention that they did several statistical analyses, but only the simple linear regression analysis is read.
- Some species lack their author and year the first time they are mentioned.
- Subjective adjectives should be avoided.

Author Response
Reviewer 1
The authors acknowledge the careful and valuable comments of all the Reviewers. We regret that the wrong version of the file was uploaded for the revision, which omitted Tables 2, 3, and 4 (quoted in the results and discussion of the work. The tables contain an important summary of field research and literature review, as well as a critical debate on endemic and alien species of ladybirds on the studied islands. The lack of tables could be an important reason for critical comments from reviewers.The authors accepted all corrections and most of the suggestions, and improved some fragments in response to some other comments. Below is the detailed list of all revisions undertaken, and those that were not implemented.(Reviewer’s remarks are highlighted in grey)
-------------------------
Comments and Suggestions for Authors
- The manuscript is interesting, as is the study question... however the experimental design is deficient and poorly planned. – The authors disagree with this remark: the field surveys were planned on different dates (mostly about one week-long field visits) that represent various seasons suitable for observing insects of the Canary Islands. This approach yielded positive results during our earlier surveys of the remaining Canary Islands (Romanowski et al. 2019, 2020a, 2020b, 2020c, 2023). Field surveys were combined with literature data to analyze the ladybird beetle faunas of the two islands
- Sampling was carried out in an irregular, non-systematized manner, which is why the data from both islands cannot be compared with each other. – The authors disagree with this remark: the field surveys were carried out in various seasons (summer excluded) suitable for observing insects of the Canary Islands. To analyze species richness at two islands data collected during the field surveys were supplemented with complete literature data for Tenerife, La Gomera, and the whole archipelago. For details see Table 2 (missing in the previously uploaded file).
- Nor is an estimate of collecting efforts presented, nor a species accumulation curve to estimate how many species could inhabit each island. – It is difficult to define precisely the collection effort during field sampling that could be influenced by the weather, availability of habitats and specific plants, and numbers of beetles as well. In our opinion, the number of identified ladybird individuals is the best indicator of sample size and we provided these numbers in the introductory section of results. The collection effort measured in sampling days and catch varied between the islands, it was higher for Tenerife (36 days, 1560 ladybird individuals) compared to La Gomera (24 days, 808 ladybird individuals). The variation in collection efforts reflects differences in Island size and the number of habitat types between Tenerife and La Gomera.
It is not clear what accumulation curve is expected. The analysis is conducted on a combination of records collected in the period of ca 160 years (starting with pioneering studies of Wollaston 1864, 1865). For details see Table 2 (missing in the previously uploaded file).
- With the methodology they followed, at most an updated faunal checklist can be presented. – The authors disagree with this remark. While the current field surveys are presented as the “faunal checklist”, the following analysis and discussion incorporate complete records collected in the Canary Islands in the period of ca 160 years, critically reviewed by authors. For details see Tables 2,3, and 4 (missing in the previously uploaded file).
- All results and their graphs should be placed in the corresponding section, not until the Discussion. – The authors disagree with this remark as the analysis and the graph is based on a combination of the authors’ data and the results of earlier studies cited in the discussion section. For details see Tables 2,3, and 4 (missing in the previously uploaded file).
- Some species are mentioned as being of "special interest", but they do not explain why? – the authors meant species that were new to the islands studied, and that were not illustrated in our earlier publications (Romanowski et al. 2019, 2020a, 2020b, 2020c, 2023). According to the remark we removed „of special interest” (page 3).
- They mention that they did several statistical analyses, but only the simple linear regression analysis is read. – According to the remark the explanation is added in the text: „The data were analyzed for normality using the Shapiro–Wilk test.”
- Some species lack their author and year the first time they are mentioned. – According to the remark we added author and the year of the original description of all the six species mentioned for the first time in the text.
- Subjective adjectives should be avoided. – According to the remark we removed adjectives: special (page 3), Strikingly (page 12).
Reviewer1 comments marked on the pdf file:
Page 2: As long as systematic sampling is carried out over at least one year. Otherwise, the only result to be published would be a faunal list. – This comment addresses the authors’ statement that Tenerife and La Gomera may be used as a suitable model to test one of the central assumptions of the theory of island biogeography, the species-area relationship, according to which the species richness in an island is positively related to the island’s size. We are aware of the limitation of the short time of sampling and the different numbers of ladybird records collected from the two islands in the current field study. For this reason, the comparisons between the two Islands surveyed and the species-area relationship were tested on the complete list of species recorded from the Canary Archipelago during a long history of research (including our results). We are convinced the analysis was justified. A similar relationship was tested for many animal taxa with various results, e.g. positive for the Canary Islands praying mantises (Mantodea) (Wieland et al. 2014). For details see Tables 2,3, and 4 (missing in the previously uploaded file)
Page 3: The sampling was not systematic, it was done on dates that seemed random. – The sampling was done on different dates (mostly about one week-long field visits) that represent various seasons suitable for observing insects of the Canary Islands. This approach yielded positive results during our earlier surveys of the remaining Canary Islands (Romanowski et al. 2019 , 2020a, 2020b, 2020c, 2023)
Page 3: What was the collection effort on each island? Was it similar? This is crucial to be able to compare the diversity between both islands. – It is difficult to define precisely the collection effort during field sampling that could be influenced by the weather, availability of habitats and specific plants, and the number of beetles as well. In our opinion, the number of identified ladybird individuals is the best indicator of sample size, and we provided these numbers in the introductory section of the results. The collection effort measured in sampling days and catch varied between the islands, it was higher for Tenerife (36 days, 1560 ladybird individuals) compared to La Gomera (24 days, 808 ladybird individuals). The variation in collection efforts reflects differences in island size and the number of habitat types between Tenerife and La Gomera.
Page 3: What do you mean by special interest? – we meant species that were new to the islands studied, and that were not illustrated in our earlier publications (Romanowski et al. 2019, 2020a, 2020b, 2020c, 2023). According to the remark we removed „of special interest” (page 3).
Page 3: This graph should be presented in the results, not in the discussion. – not implemented, the authors disagree with this remark. The graph is based on a combination of the authors’ data and the results of earlier studies cited in the discussion section and the Table 2 (missing in the previously uploaded file).
Page 3: All? What are the others?... only linear regression is mentioned here – . According to the remark the explanation is added in the text (page 3): „The data were analyzed for normality using the Shapiro–Wilk test.”
Page 3: It is essential to develop a species accumulation curve to calculate the estimated species. – It is not clear what accumulation curve is expected. The analysis is conducted on a combination of records collected in the period of ca 160 years (starting with pioneering studies of Wollaston 1864, 1865). For details see Table 2 (missing in the previously uploaded file).
Page 13: add author, year –(6 comments) – According to the remark we added author and the year of the original description of all the six species mentioned for the first time in the text.
Page 13: avoid subjective adjectives – According to the remark we removed adjectives: special (page 3), Strikingly (page 12).
Page 14: Graphs must be presented in the results. – not implemented, see the response above. Please note that according to the remark of Reviewer 3 we added a new graph in the results.
Page 14: mention P value – According to the remark we added „p=0.03”
- 16 in blue – not implemented (will be published as „Cross Ref”)

Reviewer 2 Report
Comments and Suggestions for Authors
With their manuscript, Romanowski et al. contribute to the knowledge of the Coccinellidae fauna of two Canarian islands, Tenerife and La Gomera. The work is part of a larger effort to inventory the ladybird fauna of the Canaries and builds on a considerable effort. Whilst I praise this effort and acknowledge the importance of such species lists, I have two major issues with this work:
1. In the 21st century, similar inventories only have their merits if the data are compliant with the findability, accessibility, interoperability, and reusability (FAIR) principles. For this, publishing a species and location list is not sufficient and therefore I strongly suggest the authors prepare their data in a machine-readable format as well and publish them on openly available biodiversity repositories, with GBIF being the best option. Without doing so, I cannot recommend the publication of this article because it has too little merit.
2. Using MacArthur’s and Wilson’s theory on island biogeography in this form is erroneous and both the analysis and interpretation are invalid.
a. Island area may be influential but, in the case of Tenerife which is the most visited island in Canaries, it is clear that inbound traffic is just as influential as island size would be as predicted from the theory of island biogeography. Indeed, whether the theory stands for coccinellids should be tested on species that arrived on the island through natural dispersal processes (or evolved there) and those whose dispersal was facilitated by human movements (particularly tourism and freight) should be excluded.
b. Fitting a line with seven data points is not highly convincing, particularly if one point is clearly outside of 95% confidence intervals and both Lanzarote and La Palma are very close to that. Again, considering human visitations from tourism would be key in this aspect and when discussing current species numbers that and not the theory of island biogeography should be highlighted.
Minor comments:
1. Since line numbering is missing, adding specific comments to the text is difficult. Therefore, I will only give some more general notes. However, for the next version (if any) the authors should add line numbering.
2. In the Introduction, the differences between “endemics”, “old inhabitants”, “newcomers” and “alien species” should be defined with particular attention to “old inhabitants”, “newcomers”.
3. The Methods section lists a simple linear regression as statistical analysis but it does not claim whether the assumptions for using this method met. Moreover, the results of this analysis are not presented in the Results section, only in the Discussion.
4. No identification keys are listed among the references used for identifying the specimens.
5. In this form, Table 1. has very little merit. In my opinion, showing sampling locations on a map would be a better way of representing them, and, importantly, for each sampling location the overall species number collected there, as well as the number of indigenous (endemics + native but not endemic) and alien species, should be provided. Summary values of these should also be provided for both islands separately. Since the number of indigenous species should be the basis for investigating MacArthur’s theory, these numbers are highly important.
6. Additional summary tables/figures showing subfamily representations, or feeding guilds would also be interesting.
7. English needs some attention, there are several grammatical errors.
Comments on the Quality of English LanguageThe text is generally understandable but English still needs some attention, there are several grammatical and stylistic errors.
Author Response
The authors acknowledge the careful and valuable comments of all the Reviewers. We regret that the wrong version of the file was uploaded for the revision, which omitted Tables 2, 3, and 4 (quoted in the results and discussion of the work. The tables contain an important summary of field research and literature review, as well as a critical debate on endemic and alien species of ladybirds on the studied islands. The lack of tables could be an important reason for critical comments from reviewers.The authors accepted all corrections and most of the suggestions, and improved some fragments in response to some other comments. Below is the detailed list of all revisions undertaken, and those that were not implemented.(Reviewer’s remarks are highlighted in grey)
With their manuscript, Romanowski et al. contribute to the knowledge of the Coccinellidae fauna of two Canarian islands, Tenerife and La Gomera. The work is part of a larger effort to inventory the ladybird fauna of the Canaries and builds on a considerable effort. Whilst I praise this effort and acknowledge the importance of such species lists, I have two major issues with this work:
- In the 21st century, similar inventories only have their merits if the data are compliant with the findability, accessibility, interoperability, and reusability (FAIR) principles. For this, publishing a species and location list is not sufficient and therefore I strongly suggest the authors prepare their data in a machine-readable format as well and publish them on openly available biodiversity repositories, with GBIF being the best option. Without doing so, I cannot recommend the publication of this article because it has too little merit. – In response to this very categorical and firm statement of the reviewer, the authors wish to indicate that this paper constitutes more than just a species list convenient for deposition in available biodiversity repositories. Similarly to our earlier assessment of the coccinellids of the Canary Islands (Romanowski et al. 2019, 2020a, 2020b, 2020c, 2023) we aimed at critical analysis of historical records and assessing the status of endemic species. For details see Table 2 (missing in the previously uploaded file. This analysis leads to taxonomic decisions, reveals alien species, and enable to asses the species richness. In this paper, we additionally analyze the importance of Tenerife for the diversification and dispersal of ladybirds within the archipelago. We also support FAIR principles and make all the data collected (including location coordinates, dates, and additional details) fully available for the readers. The rapid development of AI applications will most probably result in the automatic inclusion of all faunistic data, including this data set, into biodiversity repositories sooner than expected. To sum up: we are not going to implement this comment.
- Using MacArthur’s and Wilson’s theory on island biogeography in this form is erroneous and both the analysis and interpretation are invalid. – the authors disagree that analysis was erroneous, see below and Tables 2,3, and 4 (missing in the previously uploaded file).
- a. Island area may be influential but, in the case of Tenerife which is the most visited island in Canaries, it is clear that inbound traffic is just as influential as island size would be as predicted from the theory of island biogeography. Indeed, whether the theory stands for coccinellids should be tested on species that arrived on the island through natural dispersal processes (or evolved there) and those whose dispersal was facilitated by human movements (particularly tourism and freight) should be excluded. – We are aware of the influence of incoming tourists and cargo on the colonisation of oceanic islands, that may play increasing role for two central Canary islands , especially Tenerife. We tested the size-species richness relationship for all main Canary Islands - a similar relationship was tested for many animal taxa with various results, e.g. positive for the Canary Islands praying mantises (Mantodea) (Wieland et al. 2014). In the discussion we indicate that Tenerife is the most susceptible to colonization by non-native ladybirds, and in the last sentence of the discussion we wrote „ The importance of Tenerife as a major recipient of alien species in the Canary Islands is likely to be reinforced by intensive tourism and trade-related transport.”:
- Fitting a line with seven data points is not highly convincing, particularly if one point is clearly outside of 95% confidence intervals and both Lanzarote and La Palma are very close to that. Again, considering human visitations from tourism would be key in this aspect and when discussing current species numbers that and not the theory of island biogeography should be highlighted. – we agree with the remark, in the last sentence of the discussion we wrote „ The importance of Tenerife as a major recipient of alien species in the Canary Islands is likely to be reinforced by intensive tourism and trade-related transport.”
Minor comments:
- Since line numbering is missing, adding specific comments to the text is difficult. Therefore, I will only give some more general notes. However, for the next version (if any) the authors should add line numbering. – we used the template provided on the Insects web page (without line numbering). According to the remark we will attach the additional file with line numbering.
- In the Introduction, the differences between “endemics”, “old inhabitants”, “newcomers” and “alien species” should be defined with particular attention to “old inhabitants”, “newcomers”. – According to the remark we rephrased the sentence to: … Canarian and Macaronesian endemics, species common to the Canary Islands and neighbouring areas of NW Africa and/or SW Europe or alien species from remote regions…”
- The Methods section lists a simple linear regression as statistical analysis but it does not claim whether the assumptions for using this method met. – According to the remark we added the information in the method section that data were checked for normality “The data were analyzed for normality using the Shapiro–Wilk test.”
Moreover, the results of this analysis are not presented in the Results section, only in the Discussion. – The authors disagree with this remark as the analysis and the graph is based on a combination of the authors’ data and the results of earlier studies cited in the discussion section. For details see Table 2 (missing in the previously uploaded file.
- No identification keys are listed among the references used for identifying the specimens. – We did not use any identification keys, firstly because there are no such keys for the Canary Islands Coccinellidae, and secondly because keys always have limited reliability. We dissect and examine the genitalia of any specimen that is not easily identifiable. In doubtful cases (e.g. no males collected), we identify specimens to the genus level. Sometimes, if necessary and the material is available, we examine type specimens (see e.g. our Fuerteventura paper (Romanowski et al. 2019))
- In this form, Table 1. has very little merit. In my opinion, showing sampling locations on a map would be a better way of representing them, and, importantly, for each sampling location the overall species number collected there, as well as the number of indigenous (endemics + native but not endemic) and alien species, should be provided. Summary values of these should also be provided for both islands separately. Since the number of indigenous species should be the basis for investigating MacArthur’s theory, these numbers are highly important. –Tabularized data, but not maps, can provide detailed coordinates that can be reused by other researchers, in biodiversity repositories etc. For details see Tables 2, 3 and 4 (missing in the previously uploaded file.
- 6. Additional summary tables/figures showing subfamily representations, or feeding guilds would also be interesting. – We regret that the wrong version of the file was uploaded for the revision, which omitted Tables 2, 3, and 4 (quoted in the results and discussion of the work). We added the graph 4 in the last paragraph of the results.
- English needs some attention, there are several grammatical errors. – According to the remark we corrected errors and are ready to use mdpi language service if the revised paper is accepted for publication
Comments on the Quality of English Language. The text is generally understandable but English still needs some attention, there are several grammatical and stylistic errors. – According to the remark we corrected errors and are ready to use mdpi language service if the revised paper is accepted for publication

Reviewer 3 Report
Comments and Suggestions for Authors
Dear authors,
You have conducted an intensive and i presume exhausting filed work, and collected a great amount of interesting data. The manuscript is well written, but some parts need to be more clearly presented, and/or expanded. It would contribute significantly to the manuscript if you could calculate at least one diversity index such as Shannons, Simpsons, or any other. Although you talk about diversity regarding the species composition and endemics and aliens, the manuscript mostly refers only to the species richness (one factor of diversity). Abundance of the species, or their diversity (expressed by a diversity, dominance, or evenness) is nowhere mentioned. You have that data so try expanding the manuscript with results and discussion on the abundance, diversity, dominance or evenness. A table with the listed species names and present/absent data for each island should be added at least as a supplementary file to clearly present the obtained species diversity data, as this is not purely a faunistic paper according to the title. When listing the findings of the species, some data needs to be specified more precisely. The point-by-point comments are in the attached file.
All the best,
Reviewer

Author Response
The authors acknowledge the careful and valuable comments of all the Reviewers. We regret that the wrong version of the file was uploaded for the revision, which omitted Tables 2, 3, and 4 (quoted in the results and discussion of the work. The tables contain an important summary of field research and literature review, as well as a critical debate on endemic and alien species of ladybirds on the studied islands. The lack of tables could be an important reason for critical comments from reviewers.The authors accepted all corrections and most of the suggestions, and improved some fragments in response to some other comments. Below is the detailed list of all revisions undertaken, and those that were not implemented.(Reviewer’s remarks are highlighted in grey)
Dear authors,
You have conducted an intensive and i presume exhausting filed work, and collected a great amount of interesting data. The manuscript is well written, but some parts need to be more clearly presented, and/or expanded. It would contribute significantly to the manuscript if you could calculate at least one diversity index such as Shannons, Simpsons, or any other. Although you talk about diversity regarding the species composition and endemics and aliens, the manuscript mostly refers only to the species richness (one factor of diversity). Abundance of the species, or their diversity (expressed by a diversity, dominance, or evenness) is nowhere mentioned. You have that data so try expanding the manuscript with results and discussion on the abundance, diversity, dominance or evenness. – According to the remark we calculated the dominance for Tenerife and La Gomera ladybird communities and added the graph 4 in the last paragraph of the results.
A table with the listed species names and present/absent data for each island should be added at least as a supplementary file to clearly present the obtained species diversity data, as this is not purely a faunistic paper according to the title. – We regret that the wrong version of the file was uploaded for the revision, which omitted Tables 2, 3, and 4 (quoted in the results and discussion of the work.
When listing the findings of the species, some data needs to be specified more precisely. The point-by-point comments are in the attached file. – We are afraid the file with the comments was not attached for our attention.

Reviewer 4 Report
Comments and Suggestions for Authors
Dear Authors,
I read carefully your submitted mns Insects-3029040, that I consider a nice and interesting study on ladybirds , Coleoptera Coccinellidae of the Canary Islands, Tenerife and La Gomera. Data recorded are important for these beetles :native and alien species, also new species identified, host plant range and diversity, relationships with hosts and habitats colonized. However in my review, see please the attached word file of the text, I noticed several lacks of data regarding the Introduction (omitted references), the Material and Methods ( omitted names of host plants, the morphological identification of ladybirds species and the lack of morphological and anatomical details), the Results (lack of developmental stages referring to the collected specimens and names of hosts of several beetles species). I suggest to re-write the M&M chapter (clarify in particular the two methods used to classify the species, morphological and phylogenetic) and to add the missing details into species list. These data are important to understand the differences of diversity in the two studied islands. Figures and Table are clear and easy to read, only the caption of table 3 has to be summarized and the period , lines 535-544, can be included into Discussion. This chapter includes several critical comments referring to the native , tha alien species and new ones , and the relationships with the hosts living in the studied areas. I suggest to consider my review notes and suggestions because they could improve the quality presentation of the mns and its scientific soundness.
Sincerely

I suggest to re-write M&M section and to review the Discussion (just "polishing" for English language) with the help of a mother language lecturer.
This revision could improve the quality presentation of the text and its scientific soundness.
Author Response
With the co-authors we acknowledge valuable comments of the Reviewer. The authors accepted all corrections and most of the suggestions, and improved some fragments in response. The detailed list of all revisions undertaken, and our explanations is provided below (Reviewer’s remarks are highlighted in grey).
Response to the reviewers' comments
Reviewer 4
Dear Authors,
I read carefully your submitted mns Insects-3029040, that I consider a nice and interesting study on ladybirds , Coleoptera Coccinellidae of the Canary Islands, Tenerife and La Gomera. Data recorded are important for these beetles :native and alien species, also new species identified, host plant range and diversity, relationships with hosts and habitats colonized. However in my review, see please the attached word file of the text, I noticed several lacks of data regarding the Introduction (omitted references), the Material and Methods ( omitted names of host plants, the morphological identification of ladybirds species and the lack of morphological and anatomical details), the Results (lack of developmental stages referring to the collected specimens and names of hosts of several beetles species). I suggest to re-write the M&M chapter (clarify in particular the two methods used to classify the species, morphological and phylogenetic) and to add the missing details into species list. These data are important to understand the differences of diversity in the two studied islands. Figures and Table are clear and easy to read, only the caption of table 3 has to be summarized and the period , lines 535-544, can be included into Discussion. This chapter includes several critical comments referring to the native , tha alien species and new ones , and the relationships with the hosts living in the studied areas. I suggest to consider my review notes and suggestions because they could improve the quality presentation of the mns and its scientific soundness.
Sincerely
Page 2: Please, record a reference concerns to the native species. – The proper reference [1 – suppl. material] is given in the original text, with the indication that the data are contained in the supplementary material.
Page 2: Cite a proper Literature reference. – According to the remark we added the citation [16].
Page 2:The note could be better recorded in the results. – According to the remark we transferred the note into the Results section.
Page 3: I suggest to cite such ornamental plants, because they are the vegetable hosts of ladbird preys. – According to the remark we cited the names of specific ornamental plants from which ladybirds were collected in the Results section.
Page 3: Please, record the habitats and the plants investigated for ladybirds. These relationships can suggest interesting comments into Discussion. – According to the remark we added the information that Ladybirds were collected in all the habitats described above (in lines 95-104).
Page 3: Please, explain if the species were identified by morphological characters and record the literature reference followed. – According to the remark we added “The species were identified by morphological and anatomical details documented in our earlier papers [11, 13-15].
Page 3: suggest to explain if the systematic order used to classify the species is based on phylogenetic analysis made by CHE et al .2021, the reference cited. – According to the remark we supplemented the sentence with “… phylogenetic analysis made by…”
Page 3: Which ones? Please, these characters have to be recorded and also the species identified through them. – We do not report these details here to avoid repetitions and to keep the paper compacted. All the morphological and anatomical details of the species mentioned are described and illustrated in result section (including figures 2,3, and 4).
Page 4: I suggest to record the specimens collected with their developmental stage (i.e. 2 adults or 2 larvae and 1 pupa, etc.). These because the finding of young specimens indicates that the plant is really the host plant (where the species breads) and not a simple collecting site. These are important biological data for each species. – We agree with the remark. In fact we provided in the text all recorded larvae and pupa in the description of individual species. According to the remark and to clarify the presentation we mofified the fragment marked: “… number of adults, number of larvae and pupa (when recorded), …”
Page 5: Please record the name of such ornamental plants. – According to the remark we replaced “collected from ornamental plants” with „collected mostly from N. oleander, Hibiscus sp.”
Page 5: Grasses? . – No, kind of unidentified annual flowering plant. We can not provide more details and suggest to use “herbaceous plants”.
Page 8: Please, record the name of plant. – We added: … collected from N. oleander and unidentified ornamental plant…”
Page 12: On which plant? – We added “from Laurus sp.”
Page 13: The last two species are native or introduced species? Specify… – the two species are endemic, as indicated in the original word file.
Page 18: The caption of Table 3 is too big (line 535-544) and can be added to the Discussion. – According to the remark we modified the caption by transferring one fragment (“The ‘near-endemic’ category including two species (Parexochomus quadriplagiatus and Tetrabrachys deserticola) that, in addition to the Canary Islands, have also been reported from NW Africa (Morocco) [30]”) to line 532 of Discussion, and rewriting the text:
Table 3. Species of Coccinellidae recorded on individual islands of the Canary Islands that are considered endemic or near-endemic to Macaronesia. Two species previously thought to be endemic to the Canary Islands [2,13] were not included: Nephus incisus (Lindberg), which appeared to have a much wider geographical range [49], and Novius conicollis Korschefsky, which was synonymized with N. cruentatus (Mulsant) [15]. Data on the distribution of Diomus gillerforsi comes from the papers by Fürsch [9] and Eizaguirre [10], and this study. The distribution data for the remaining species is taken from our recent papers [11-15]. Island symbols as in Table 2.
Reviewer 5 Report
Comments and Suggestions for Authors
Coccinellid occurrence in Canary Islands with particular focus on comparison of two islands of comparable geographies but different sizes. In addition to recording occurrence and providing information detailing previous records from the islands, the outcome supports the theory that more species (greater diversity) are found on larger islands compared to smaller islands.
A really nice paper reporting on an impressive data set demonstrating attention to geography of islands and diverse potential variables.
P6 L 200 re Harmonia axyridis at present there is only one recordfrom Australia (https://biocache.ala.org.au/occurrence/search?q=lsid%3AALA_DR22913_1610&qualityProfile=ALA&offset=80&max=20) and it is on the Invasive Species Watchlist (https://invasives.org.au/wp-content/uploads/2019/06/Invasion-Watch_Harlequin-ladybird.pdf)
Author Response
With the co-authors we acknowledge valuable comments of the Reviewer. The detailed list of all revisions undertaken, and our explanations is provided below (Reviewer’s remarks are highlighted in grey).
Response to the reviewers' comments
Reviewer 5
Coccinellid occurrence in Canary Islands with particular focus on comparison of two islands of comparable geographies but different sizes. In addition to recording occurrence and providing information detailing previous records from the islands, the outcome supports the theory that more species (greater diversity) are found on larger islands compared to smaller islands.
A really nice paper reporting on an impressive data set demonstrating attention to geography of islands and diverse potential variables.
P6 L 200 re Harmonia axyridis at present there is only one record from Australia (https://biocache.ala.org.au/occurrence/search?q=lsid%3AALA_DR22913_1610&qualityProfile=ALA&offset=80&max=20) and it is on the Invasive Species Watchlist (https://invasives.org.au/wp-content/uploads/2019/06/Invasion-Watch_Harlequin-ladybird.pdf) – We thank the reviewer for providing the link to interesting observation. To keep our paper focused and compacted we do not intend to comment the issue of establishment of H. axyridis in Australia
Round 2
Reviewer 1 Report
Comments and Suggestions for Authors
There were very few improvements to the article. The authors simply rejected most of the comments from the 3 reviewers. Although I consider that they were key comments and suggestions for the manuscript to improve to such a degree that it could be considered for acceptance and publication. For this reason I maintain my opinion of rejecting the article.
Comments on the Quality of English LanguageMinor editing of English language is required
Author Response
With the co-authors we acknowledge valuable comments of the Reviewers. The authors accepted all corrections and most of the suggestions, and improved some fragments in response. The detailed list of all revisions undertaken, and our explanations is provided in the response to Reviwer 4, and Reviwer 5
Reviewer 4 Report
Comments and Suggestions for Authors
Dear Authors,
I read carefully the new revised version of the mns Insects-3029040 you proposed, following the notes/suggestions of the reviewers (see, please, the attached pdf file). I appreciated your efforts and read your replies to my suggestions: I realized that you have been agreed and produced review modifications to the text (the citing of references, the names of host plants of the ladybird collecting sites, etc. as suggested). These addictions have increased the scientific soundness and quality presentation of the text, and favour the interest to the readers.
Sincerely
